# OpenReview forum: "Is it safe to share your files? An Empirical Security Analysis of Google Workspace"
_ACM.org/TheWebConf/2024/Conference — TheWebConf24 Oral_

### Official Review · Reviewer_J5sZ · 2023-10-25

**Novelty:** 5
**Technical Quality:** 5

**Review:**

**Strengths**

+ Large-scale measurement study (4,732 add-ons)
+ Security risks for Google Workspace
+ Seemingly interesting results

**Weaknesses**

- Data collection methodology/setting missing
- Lack of experimental evidence for generality claims
- Pure manual analysis with no automation
- Threat model is a bit strong
- Adherence to responsible disclosure standards is unknown

Thank you for submitting your work to WebConf'24. The security of BCPs certainly stands as an imperative cornerstone in the modern landscape of organizational risk management and resilience, as vulnerabilities in these systems can expose sensitive data and assets to potential threats and malicious actors.

The main strength of this paper lies in the large-scale measurement and the corresponding findings of the analysis. In particular, the results are surprising to the extent that the authors identify security risks in over 70% of the tested add-ons, and follow (some of) these all the way to exploits.

Another strength of this work is the threat modeling analysis performed by the authors and the proposed security risks/ attacks for Google Workspace, which could be enlightening.

At the same time, the paper comes with a couple of rather significant weaknesses:

- Firstly, the work largely follows manual inspection for security analysis which is error-prone and hard to reproduce (while I appreciate the huge effort spent to conduct this study). This also directly counters the motivation in the intro that better program analysis tools is needed for BCPs. I expected the authors to have at least some form of automation, which can benefit our community.

- Second, the paper lacks experimental evidence to substantiate its broad assertions regarding BCPs. While the paper exclusively investigates Google Workspace, it makes generalizations that extend to all BCPs (e.g., Sections 1 to 3). The authors should consider conducting experiments on other platforms or, alternatively, limit their claims to be more specific to Google Workspace. For example, it is unclear if the proposed attacks in Table 3 transfer to other platforms, or if the proposed threat model and attacker capabilities of Section 3 still hold for alternative platforms.

- Third, the threat model is a bit strong and narrow, as malicious users can only install maliciously crafted add-ons on "their own" BCPs without the vetting process of Google. Also, the paper should specify more explicitly what are the exact set of assumptions it makes about attacker capabilities and to what extent they are realisitic.


**Presentation:**

In general, the paper has an overally good presentation and writing quality. However, here are some nitpicks:

> Traditional security methods like static code analysis and dynamic
injection execution, which work in other scenarios, are ineffective
in BCPs.

- This is a controverisal claim without supporting evidence. I cannot think of any reason as to why common program analysis techniques like SAST/DAST should not work here. I agree that if the code is unavailable, static analysis is infeasible. But what about dynamic testing? It would be great if the text can better substantiate these claims.

- Section 3.1 is confusing in the sense that it uses the term "user" interchangebly for both benign users and attackers.


## Update After Rebuttal

Thank you for answering my questions. The rebuttal sufficiently addresses my (main) concerns and I am happy to recommend this work for acceptance.

**Questions:**

- Data collection: how did you obtained the dataset of add-ons for the manual analysis? What was the followed methodology to select these 4,732 add-ones and how did you collect their (client-side) source code? I could not find these details in the paper, which are important for reproducibility.

- In Section 7, the claim that more than 70% of the add-ons are vulnerable implies that you created exploits for all of them. Is this really the case? I suspect that these 70% may primarily align with your indicator oracle for security risk-associated behaviors, suggesting that the paper is not using the most proper terminology (I stand corrected if I am wrong).

- As for vulnerability disclosure, the paper only notifies Google. But how about other benign add-ons that are vulnerable and part of the 70% that are vulnerable? The authors should clarify compliance with standards of responsible disclosure.

- Question about vulnerability confirmation: did Google confirmed the reported behaviour as a vulnerability? What was the CVSS score? Did they decided to patch? I am curious to know the answer, and I believe documenting as much of that process as possible in the paper (especially regarding the type of the defense/patch applied) would be very insightful for other researchers.

**Ethics Review Description:**

Adherence to responsible disclosure standards is unknown: the paper finds that 70% of the add-ons in Google workspace are vulnerable to a series of attacks the authors propose. However, vulnerability disclosure has been done purely to Google, missing other benign add-ons that are vulnerable and without a clear disclosure plan for other parties.

**Reviewer Confidence:**

3: The reviewer is confident but not certain that the evaluation is correct

**Scope:**

4: The work is relevant to the Web and to the track, and is of broad interest to the community

---

### Official Review · Reviewer_LzmV · 2023-11-21

**Novelty:** 5
**Technical Quality:** 4

**Review:**

This paper studies the security issues of the business collaboration platforms (BCPs), i.e., Google Workspace, which allows users to share various file resources and write custom add-on applications to facilitate collaborations among other working partners. Through a security analysis, three vulnerabilities were discovered due to the improper design of access control. These vulnerabilities can be exploited for a number of attacks to conceal file metadata, hijack controls, and leak sensitive user data.

Strengths
- Discovered three types of new vulnerabilities in Google Workspace
- Demonstrated practical attacks exploiting the vulnerabilities
- Measurement study to understand the problem from a market scale
- Discussion of root cause, countermeasures, and responsible disclosure

Weaknesses
- Unclear details in the measurement study
- No discussion on other BCPs other than Google

Overall, I am positive about this paper, as it is the first study to discover a few new vulnerabilities in Google GBP and has demonstrated a number of practical attacks via malicious add-on applications. The threat model makes sense and the security implications seem to be significant as the attacks could escalate the attacker’s privilege and steal very sensitive data. Nevertheless, I still have some suggestions to improve the paper described below.

The description of the measurement study is not crystally clear. I am particularly confused by why existing applications are subject to the uncovered vulnerabilities. If I understand it correctly, the attack is mainly achieved via an attacker-crafted malicious add-on application. Therefore, what do the measurement results try to convey? For instance, it is mentioned that “14% of the add-ons exhibit the potential to initiate a worm attack”. Does this indicate that these 14% add-ons could be potentially malicious, i.e., malware on the market? The methodology used to conduct the measurement is also unclear. I suppose it was done by scanning a predefined set of APIs (that could be abused by attackers) in the add-on’s application code.

The paper also does not discuss other BCPs other than Google, such as Microsoft’s BCP. While I don’t think the authors should also study that in this paper, having a discussion on whether other BCPs could suffer from similar issues would make the paper stronger.

**Questions:**

- Can any users develop custom add-ons and publish them on the Google Workspace market? Does Google have a vetting process for add-on developers and the applications on the market?
- How many add-on applications are on Google’s market?

**Reviewer Confidence:**

3: The reviewer is confident but not certain that the evaluation is correct

**Scope:**

3: The work is somewhat relevant to the Web and to the track, and is of narrow interest to a sub-community

---

### Official Review · Reviewer_sqE1 · 2023-11-22

**Novelty:** 5
**Technical Quality:** 5

**Review:**

The paper identifies three vulnerabilities due to permission inconsistencies in Google Workspace add-ons.

The paper has potential to identify interesting issues arising from the two types of permission models (user and add-on) in Google Workspace. While the paper already make progress towards identifying these issues, the descriptions and analysis are sometimes inadequate to fully grasp the impact of these issues.
I do like the clear listing of the three vulnerability types, it helps to connect them to the attacks. I think a reference to Vulnerability 2 is still warranted somewhere in Section 5.
To make a stronger case, I think the attack model could make the "insider threat" attacker more explicit.

The attack descriptions lack some technical details and reflections.
* S4: how can attackers use the stolen information for phishing? Is it realistic that the workspace is the only place where they can obtain this information (especially as an insider)?
* S4: _"Google restricts some features"_ -- which ones?
* S4, bullet 2: "without raising any alert" -- is this complete invisible? wouldn't users get a notification?
* S4: how does knowledge relate to access?
* S5: _"We only discuss one type of
malicious code shown in Figure 5"_ -- which others exists? _"the attacker can spoil more by
inserting other malicious code along with this worm attack."_ -- how?
* S6: _"Our experiment demonstrates that, in certain cases, even if the owner of the resource did
not install the malicious add-on, the attacker can still exploit vulnerabilities to gain the privilege of removing all collaborators from
the resource, leaving only the owner with access."_ -- how?

It is also not clear whether all analyses were conducted systematically. For example, in Section 4, was the fact that _"The resource participant who joined under general mode cannot
view the “version history” or “collaborators’ account” of the file
compared with the participant who joined under restricted mode."_ the result of a systematic search, or were these attributes discovered serendipitously? Are there more of such attributes?

Unfortunately, the writing is not always very clear, which does not help to properly understand your contributions. As one example, I found the bullet on "App-to-App Control Hijacking Attack" in the introduction very difficult to parse. Another example: on page 5, "steeled" should be "stolen". I would suggest a detailed readthrough of the text to make sure everything is clear.

There are also a few omissions in Section 6.2: I cannot find any "definition in Section 3.2", and Figure 7 seems to wrongly referenced.

I appreciate the summary provided by the root cause analysis. However, the proposed countermeasures feel somewhat superficial, especially the invariants: is this not just searching for _any_ string constants?

In terms of novelty, to me the most closely related work appears to be reference [19]. I think this paper is sufficiently different, covering a different domain (workspaces vs. messaging) and uses sufficiently different methods. However, it would be helpful if this is made more clear in the paper itself as well, through an explicit comparison with the related work.

It might have been interesting to compare these add-ons to the more well-studied spaces of browser extensions and mobile apps, in terms of their permission models and the vulnerabilities discovered within them.

Smaller comments:
- The abstract makes exaggerated claims ("alarming discovery"...).
- Table 1: should the listing for Gmail have two `mail` permissions, or should one of the two be `mail.readonly`?
- Figure 5 feels unnecessary for me, all relevant details are already in the text. This could be used to gain space. (Figures 6 and 7 do have added value over the text.)

**I have read the rebuttal.**

**Questions:**

Please see my review for questions related to the attack description and the systematic approach to the analysis.

**Reviewer Confidence:**

2: The reviewer is willing to defend the evaluation, but it is likely that the reviewer did not understand parts of the paper

**Scope:**

3: The work is somewhat relevant to the Web and to the track, and is of narrow interest to a sub-community

---

### Official Review · Reviewer_ECuH · 2023-11-23

**Novelty:** 6
**Technical Quality:** 5

**Review:**

This paper conduct the first study on the effectiveness of the cross-entity resource management in Google Workspace, and they spent much maunal efforts
to collect the data and analyze the results, which is very impressive.
The paper is well written and the results are interesting, although there are some minor issues that need to be addressed.

- Line 516, "... from Doc to Slide." -> "... from Doc to Sheet."
- Line 561, "To be detailed, First, ..." -> "To be detailed, first, ...".
- Related Work section, ref [44] and [48] are referenced using the first name of the first author. Please use the last name of the first author instead.

**Questions:**

- The authors conducted a considerable amount of manual efforts on manual analysis, which is impressive. However, the details for the large-scale study
are not clear. I am wondering how the authors automate this analysis.

**Reviewer Confidence:**

3: The reviewer is confident but not certain that the evaluation is correct

**Scope:**

4: The work is relevant to the Web and to the track, and is of broad interest to the community

---

### Official Review · Reviewer_Atpx · 2023-11-23

**Novelty:** 5
**Technical Quality:** 6

**Review:**

**Summary**
In this paper, the authors identify three vulnerabilities in the access control mechanisms in Business Collaboration Platforms (BCPs). After detailing the vulnerabilities and their corresponding attacks, to validate their findings, they performed a large-scale analysis of Google Workspace add-ons. The results show that over 70% of the add-ons suffer from at least one vulnerability. Finally they contribute by identifying a set of mitigations to these attacks.

**Evaluation**
Overall, a good paper that should be of interest to the conference attendees.

Pros:
- The submitted article is written in good English.
- The structure of the article is appropriate and easy to follow.
- Clear presentation of the vulnerabilities and their attacks.
-The results highlight the feasibility of the attacks and the need for improving the access control mechanisms in BCPs.

Some minor improvements:
- In the introduction, the contributions are presented in both the “Our work” paragraph and the “Contributions” paragraph. I suggest merging them together.
- I’m not sure I understand the value of Figure 3.

Typos:
- Page 2, line 128: BCPs supports → BCPs support
- Page 5, line 489: For example, In Figure → For example, in Figure
- Page 5, line 561: To be detailed, First, → To be detailed, first,
- References [15] and [16] seem the same.

**Questions:**

no questions

**Ethics Review Description:**

-

**Reviewer Confidence:**

3: The reviewer is confident but not certain that the evaluation is correct

**Scope:**

4: The work is relevant to the Web and to the track, and is of broad interest to the community

---

### Decision · Program_Chairs · 2024-01-22

**Decision:**

Accept (Oral)

**Comment:**

The reviewers agreed that this paper makes a significant contribution, because it identifies new vulnerabilities in Google Workspace with clear security implications. Overall, the reviewers enjoyed the reading of this paper and were impressed by the amount of work put into this research. On the other hand, they criticized the relatively narrow application field and the extensive use of manual analysis in the vulnerability detection process.

 ---